# Can Novel Biomarkers Effectively Predict Acute Kidney Injury in Liver or Kidney Transplant Recipients?

**DOI:** 10.3390/ijms252212072

**Published:** 2024-11-10

**Authors:** Hubert Zywno, Wojciech Figiel, Michal Grat, Slawomir Nazarewski, Zbigniew Galazka, Jolanta Malyszko

**Affiliations:** 1Department of Nephrology, Dialysis, and Internal Diseases, University Clinical Centre, Medical University of Warsaw, 02-097 Warsaw, Poland; jolanta.malyszko@wum.edu.pl; 2Doctoral School of Medical University of Warsaw, Medical University of Warsaw, 02-091 Warsaw, Poland; 3Department of General, Transplant, and Liver Surgery, University Clinical Centre, Medical University of Warsaw, 02-097 Warsaw, Poland; 4Department of General, Endocrinological, and Vascular Surgery, University Clinical Centre, Medical University of Warsaw, 02-097 Warsaw, Poland

**Keywords:** acute kidney injury, liver transplantation, kidney transplantation, biomarkers, diagnostic performance

## Abstract

Acute kidney injury (AKI) constitutes a common complication associated with liver or kidney transplantation, which may significantly impact the graft condition and perioperative mortality. Current AKI diagnostic criteria based on serum creatinine (sCr) and urine output alterations are widely utilized in routine clinical practice. However, the diagnostic value of sCr may be limited by various confounding factors, including age, sex, reduced or increased muscle mass, and pre-existing chronic kidney disease (CKD). Furthermore, sCr is rather a late indicator of AKI, as its concentration tends to increase only when the severity of the injury is enough to decrease the estimated glomerular filtration rate (eGFR). Recent expertise highlights the need for novel biomarkers in post-transplantation AKI diagnosis, prediction of event-associated mortality, or evaluation of indications for renal replacement treatment (RRT). Over the last decade, the diagnostic performance of various AKI biomarkers has been assessed, among which some showed the potential to outperform sCr in AKI diagnosis. Identifying susceptible individuals, early diagnosis, and prompt intervention are crucial for successful transplantation, undisturbed graft function in long-term follow-up, and decreased mortality. However, the research on AKI biomarkers in transplantation still needs to be explored. The field lacks consistent results, rigorous study designs, and external validation. Considering the rapidly growing prevalence of CKD and cirrhosis that are associated with the transplantation at their end-stage, as well as the existing knowledge gap, the aim of this article was to provide the most up-to-date review of the studies on novel biomarkers in the diagnosis of post-transplantation AKI.

## 1. Introduction

Acute kidney injury (AKI) constitutes a common complication following liver or kidney transplantation marked by high morbidity and mortality [1]. Post-transplantation AKI has a multifactorial and unclear onset, which may be associated with pre-existing end-stage liver or renal disease, the toxicity of immunosuppressive therapy, including calcineurin inhibitors administration, or intraoperative complications such as ischemia-reperfusion injury causing post-reperfusion syndrome [2,3]. Post-transplantation AKI may be often misdiagnosed as it is sometimes preceded by an initial recovery period [4]. Therefore, AKI may significantly affect the graft condition, leading to its delayed function or even acute graft rejection [5]. Considering the rapidly growing number of liver or kidney transplant candidates, a safe and effective approach towards the early recognition of post-transplantation AKI is needed to provide a prompt intervention, prevent graft loss and subsequently improve the function of the graft in the long-term follow-up. Recent studies indicate the promising role of novel biomarkers of AKI in patients undergoing liver or renal transplantation. This concern seems essential, considering that current diagnostic criteria for AKI are still based on markers of decreased kidney function, namely, serum creatinine (sCr) concentration and urine output but not on markers of injury [6]. Therefore, delayed diagnosis is possible as the alterations in sCr are detectable only when the severity of the injury is enough to decrease the estimated glomerular filtration rate (eGFR) [7]. Moreover, various confounding factors may affect the sCr concentration, i.e., age, sex, and muscle mass [8]. Interestingly, some researchers proposed that new AKI diagnostic criteria should encompass the sCr-based AKI definitions with kidney injury biomarkers [9,10]. Indeed, selected biomarkers like cystatin C, neutrophil gelatinase-associated lipocalin (NGAL), and urinary tissue inhibitor of metalloproteinases-2/insulin-like growth factor-binding protein-7 ([TIMP2*IGFBP7]) showed a superior predictive value compared to traditional markers [7,11,12]. Other biomarkers, such as kidney injury molecule 1 (KIM-1) or liver-type fatty acid-binding protein (L-FABP), also demonstrate potential as predictive markers of AKI [11,12,13,14]. Furthermore, emerging molecules like proenkephalin (PENK), C-C motif chemokine ligand 14 (CCL14), and Dickkopf-related protein 3 (DKK3) have shown promise as effective predictors of post-transplant AKI, with comparable or improved performance, specificity, and sensitivity to other biomarkers, simultaneously outperforming creatinine-based AKI calculations [15,16,17,18]. However, the current data on PENK and DKK3 are limited, necessitating further investigation. Notably, the field of AKI biomarkers lacks consistent results, rigorous study designs, external validation, and unified biomarker cut-off values [11]. Considering the existing knowledge gap, the purpose of this article was to provide the most up-to-date review of the studies on novel biomarkers in the diagnosis of AKI following liver or kidney transplantation.

## 2. Acute Kidney Injury—Definition, Classification, Epidemiology, and Clinical Outcomes

AKI is a syndrome comprising a broad manifestation of clinical conditions which, according to the Kidney Disease Improving Global Outcomes (KDIGO) guidelines, is defined as a sudden increase in serum creatine of ≥0.3 mg/dL within 48 h, an increase in serum creatinine of ≥1.5 times baseline within the prior seven days, or a decrease in urine output of <0.5 mL/kg/h for 6 h [19]. The most widely used classification of AKI distinguishes pre-renal, intrinsic renal, and post-renal AKI [20]. Examples of clinical situations for each type of AKI are shown in Figure 1. Furthermore, the RIFLE (Risk, Injury, Failure, Loss of function, and End-stage renal disease) and AKIN (Acute Kidney Injury Network) criteria allow staging AKI, depending on its severity. They are based on the dynamics of serum creatinine concentration growth and urine output decline [21]. These criteria are a useful clinical tool in diagnosis, therapeutic management, and predicting outcomes among affected patients. Nowadays, AKI constitutes a significant health problem with its still rapidly increasing worldwide prevalence. AKI holds a myriad of symptoms that may present as a mild increase in serum creatinine concentration requiring only proper intravenous fluid supplementation but also may lead to the necessity of renal replacement therapy (RRT), chronic kidney disease, end-stage kidney disease, or multiorgan failure development, which is associated with significantly increased mortality in the hospital departments [22,23]. The risk of AKI ocurrence is often significantly increased among patients with underlying diseases such as hypertension, type 2 diabetes mellitus, hepatorenal syndrome, or even severe acute respiratory syndome coronavirus 2 (SARS-CoV-2) infection, which is often associated with both acute liver and kidney failure [24,25,26,27]. Clinical outcomes of AKI are dependent on the severity and duration of renal impairment as well as the baseline renal condition of the affected patient [23,28]. Therefore, considering that individual course of AKI may be unpredictable and associated with potentially severe outcomes or significantly decreased survival among patients, early diagnosis, and prompt targeted intervention remain crucial for achieving therapeutic success. Noteworthy, current AKI diagnostic criteria are still based on alterations in serum creatinine and urine output, both late-stage kidney damage markers. Hence, further studies focused on new potential methods of early AKI diagnosis are still in demand [6,9].

## 3. Novel Biomarkers—A Promising Alternative in Acute Kidney Injury Prediction and Diagnosis

Here, we briefly described some of the novel biomarkers of AKI, with a special focus on their diagnostic performance in clinical settings other than liver or kidney transplantation. The classification of biomarkers depending on the mechanism of action and the place of injury in the nephron is shown in Figure 2. To provide a better review of the current data, in Table 1, we summarized exemplary studies on AKI biomarkers according to the clinical setting and the outcomes of their prognostic value. We also included studies focused on metabolomic biomarkers of AKI (Table 1).

### 3.1. Neuthrophil Gelatinase-Associated Lipocalin (NGAL)

NGAL is a protein representing the lipocalin family secreted by activated neutrophils. It is widely expressed in epithelial cells of the heart, lung, tubular cells of the kidney, liver, stomach, and colon [10]. The expression of the model example of the injury biomarker is induced by different inflammatory molecules such as toll-like receptors and cytokines, including interleukins 1b, -6, -17, or -22, as well as tumor necrosis factor-alpha in response to the epithelial cell injury. Therefore, it plays a vital role as an acute-phase protein during inflammation [72,73,74]. Increased expression of NGAL is linked with an injury in the proximal or distal convoluted tubule or collecting duct of the nephron. In the proximal tubule, the baseline expression of NGAL is low. It is heavily increased after the injury, whereas, in the distal convoluted tubule, baseline expression is higher and is further enhanced after the influence of the triggering factor [75]. Numerous studies have pointed out NGAL as a promising predictive biomarker of AKI, especially in various acute states. For instance, serum and urinary NGAL levels substantially increased and were positively correlated with sCr and AKI development in critically ill patients with diagnosed sepsis, heart failure, decompensated cirrhosis, and ketoacidosis, as well as those who underwent percutaneous coronary intervention as a management of myocardial infarction [29,30,31,32,35,76]. NGAL was a significantly better predictor of AKI than sCr in those individuals, confirmed in the area under the receiver operating characteristic curve (AUROC) analysis. A large meta-analysis conducted by Pan et al., which included 110 different studies on biomarkers, showed that both urinary and serum NGAL had the highest predictive value of AKI among other biomarkers. Most included studies in this analysis involved patients admitted to the ICU or after cardiothoracic surgeries [77]. Hence, the potential use of NGAL in the early diagnosis of AKI associated with acute clinical states can be valuable due to the need for immediate intervention to improve patients’ outcomes and survival. Moreover, NGAL also appears to be a useful AKI biomarker in different underlying causes such as acute pancreatitis, liver cirrhosis, and inflammatory bowel diseases [35,78]. A prospective cohort study conducted on 213 patients with decompensated cirrhosis showed that urinary NGAL effectively predicted AKI, allowing for its better discrimination between acute tubular necrosis (ATN) AKI and non-ATN AKI than sCr. C-statistic analysis revealed a better graded association between higher urinary NGAL levels and decreased 90-day transplant-free survival than a model for the end-stage liver disease (MELD) score [35]. NGAL, however, like other biomarkers, has some limitations, including the following ones: (1) lack of universal cut-off values, which may be associated with challenging discrimination of true positives from false positive and false negative results; (2) possible impact of chronic kidney disease (CKD) or systemic inflammation during sepsis on NGAL concentrations; (3) differences in diagnostic utility, depending on the stage of AKI; (4) population heterogeneity [79,80,81,82]. Furthermore, a recent study by Frydman et al. on patients diagnosed with ST-elevation myocardial infarction showed that serum NGAL indexed with eGFR had a significantly better predictive ability of AKI than non-indexed NGAL alone [83]. Therefore, the limitations discussed here can be reduced by future studies that introduce possible modifications of measurements, such as the combination of NGAL with different clinical variables. NGAL concentrations must be interpreted with caution in patients with systemic inflammatory or autoimmune diseases.

### 3.2. Cystatin C (CysC)

Cystatin C is a nonglycosylated cysteine protease inhibitor expressed by all nucleated cells, which is effectively filtrated in the glomerulus and subsequently wholly degraded in the proximal convoluted tubule without reabsorption [84]. Therefore, like sCr, CysC is a marker of filtration. Contrary to sCr, CysC constitutes a biomarker independent of age, sex, muscle mass, or inflammation [36]. Typically, CysC is not detected in urine but may be elevated in the states associated with the tubular injury [85]. CysC-based eGFR calculations may perform better than those relying on sCr in selected populations such as elderly patients, transplant recipients, or patients with cirrhosis [8]. A meta-analysis conducted by Shahraki et al. portrayed serum and urinary Cys-C as an accurate biomarker of AKI (AUC: 0.83 and 0.85, respectively) among children with the highest sensitivity and specificity for serum CysC in the cut-off points range of 0.4–1.0 mg/L [86]. Urine and serum CysC concentrations are markedly elevated in various clinical settings, including cirrhosis [36,37,87], patients hospitalized in intensive care units (ICUs) [88,89,90], acute pancreatitis [40,41,91], or chronic obstructive pulmonary disease [92]. Along with alterations in the concentration of CysC, significantly higher AUC values, sensitivity, and specificity via AUROC analysis in comparison to those with sCr were typically observed. Contrary to these data, some studies showed that CysC does not consistently outperform sCr in the diagnostic accuracy of AKI. For instance, Asakage et al., in their recent study, reported that predictive performance of urinary CysC alone or combined with other biomarkers in critically ill ICU patients was limited. This was reflected by a low AUC value of 0.57 during 1-year follow-up, despite a significant gradual elevation of concentrations of these molecules and a positive correlation with AKI severity. The reason for this discrepancy may lie, as the authors highlighted, in the lack of measurements of biomarkers in a large group of patients and the lack of urine output recording, which may have led to AKI misdiagnosis [93]. Interestingly, a study conducted by Pei et al. compared the diagnostic performance of different novel AKI biomarkers, namely, CysC, NGAL, kidney injury molecule-1, and fibroblast growth factor-23 in a Chinese cohort with diagnosed sepsis. The results revealed that serum CysC had the highest AUC and sensitivity from all examined proteins, and the diagnostic effectiveness in AKI prediction was even more accurate when combined with sera CysC and sCr [94]. This strengthens the hypothesis that the calculations involving not only traditional methods, but also novel biomarkers may lead to better early AKI prediction and diagnosis.

### 3.3. Kidney Injury Molecule-1 (KIM-1)

Kidney Injury Molecule-1 is a transmembrane glycoprotein typically undetectable in healthy kidneys, whose expression is upregulated during an injury in the proximal convoluted tubule [95]. Some studies pictured it as a reliable predictive biomarker of AKI. Brozat et al. observed that serum levels of KIM-1 were significantly elevated in critically ill septic patients who developed AKI within 48 h and those receiving RRT later. These alterations were correlated with sCr and inflammatory parameters. In univariate and multivariate regression analyses, higher KIM-1 levels were positively correlated with the necessity of RRT, multiorgan failure as well as sepsis development in patients with AKI on the day of admission [96]. Moreover, the results of the study conducted on patients with decompensated cirrhosis showed that concentrations of urinary KIM-1 combined with urinary NGAL and serum CysC were significantly higher among individuals who developed AKI and tended to increase along with the Child-Pugh staging. The combination of these three biomarkers was characterized by the highest AUC value with the highest sensitivity and specificity in AKI prediction compared to the diagnostic value of each biomarker alone [44]. KIM-1 also showed potential in the studies involving patients who underwent nephrotoxic chemotherapy. Monitoring of urinary KIM-1 allowed for an efficient AKI prediction associated with anti-cancer treatment in the study groups [43,97]. These results support the hypothesis that measurements of urinary KIM-1 may allow for early discrimination of individuals with a high risk of drug-induced AKI development, thereby improving the overall survival rate of cancer patients. Furthermore, a recent clinical trial and meta-analysis of VA NEPHRON-D indicated urinary KIM-1 as an effective biomarker of AKI-associated hemodynamic changes in patients who received combined hypotensive therapy consisting of angiotensin-converting enzyme inhibitor (ACEI) lisinopril and angiotensin receptor blocking (ARB) losartan during 12-month follow-up. In this setting, KIM-1 allowed for phenotyping and discriminating benign changes in kidney perfusion from a true intrinsic AKI [98]. This clinical trial shows that KIM-1 may be helpful in the long-term surveillance of patients treated for chronic diseases where nephroprotection is crucial, such as hypertension, CKD, and diabetes mellitus. However, KIM-1, like other biomarkers, has some limitations that involve the possible influence of proteinuria and inflammatory diseases on its concentration and diagnostic performance [99].

### 3.4. Urinary Insulin-like Growth Factor-Binding Protein 7 and Tissue Inhibitor Metalloproteinase 2 (TIMP2*IGFBP7)

Urinary TIMP2*IGFBP7 biomarker constitutes a combination of two proteins, namely, TIMP2—an endogenous inhibitor of metalloproteinases—and IGFBP7—an inhibitor of signaling in insulin-like growth factor 1 receptors. The mechanism of action of these two proteins is related to the arrest of the cell-cycle G1-phase. It is one of the protective mechanisms following AKI, as the entrance of injured tubular cells in the S-phase of the cell cycle is associated with a more severe AKI and increased mortality [100,101]. This process takes place at the very early stage of injury. Therefore, TIMP2*IGFBP7 may act as a robust biomarker of AKI prediction, allowing for prompt diagnosis and management [102]. TIMP2*IGFBP7 is one of very few AKI biomarkers that are commercially available in laboratory assays sold under the tradename NEPHROCHECK^©^ (Astute Medical Inc., San Diego, CA, USA) with increasing usage, especially in ICU departments [103,104]. Indeed, a large multi-center observational study conducted by Kashani et al. showed that the urinary combination of TIMP2 and IGFBP7 very effectively predicts AKI in critically ill patients hospitalized due to sepsis, shock, or major traumas. Furthermore, these molecules significantly outperformed either sCr or biomarkers such as sera NGAL and CysC, as well as urinary NGAL, KIM-1, interleukin-18, and liver fatty acid-binding protein 1 [49]. These results seem to be supported by a meta-analysis performed by Liu et al., which placed TIMP2*IGFBP7 as an effective AKI predictive biomarker in the ICU and patients who underwent cardiothoracic procedures such as coronary artery bypass graft or transcatheter aortic valve implantation [104]. Furthermore, TIMP2*IGFBP7 was characterized by a high diagnostic value of AKI in patients with SARS-CoV2-associated acute respiratory distress syndrome [105,106]. Contrary to these data, an observational study focused on the AKI development in patients who underwent abdominal aortic aneurysm repair showed that urinary concentration of TIMP2*IGFBP7 neither changed significantly at baseline nor post-operatively. The overall diagnostic performance was poor for two cut-off values at different stages of AKI [107]. A recent study showed that TIMP2*IGFBP7 is also an accurate predictor of the need for RRT in sepsis-induced AKI. Moreover, combining this biomarker with the Furosemide Stress Test improves AKI’s diagnostic accuracy (AUC: 0.86 vs. 0.66 for the biomarker alone) with a high specificity [52]. These results are important since early RRT initiation in septic AKI is crucial to avoid multiorgan failure. As previously discussed, combining novel biomarkers with other clinical variables may improve the diagnostic efficacy of these molecules. The main limitations of TIMP2*IGFBP7 are decreased diagnostic values in diabetes (possibility of false-positive measurements occurrence) and relatively high cost [7,108]. However, commercial implementation of this biomarker under the NEPHROCHECK^©^ assay may constitute a positive sign for the broader adoption of TIMP2*IGFBP7 and other novel biomarkers of AKI for clinical use in the future.

### 3.5. Proenkephalin (PENK)

Proenkephalin is a protein belonging to the enkephalin peptide family and is freely filtrated in the glomerulus. This molecule is a potent agonist of delta opioid receptors, and its highest density is found in the kidney. The functions of PENK in the human body have yet to be fully discovered. However, it is assumed that this protein may induce diuresis and natriuresis via opioid receptor agonism or inhibition of antidiuretic hormone [16]. PENK is purely filtrated by the glomerulus and is not reabsorbed. However, contrary to sCr, the PENK concentrations are independent of age and sex [16,109]. A recent meta-analysis of 11 observational studies conducted by Lin et al. showed that PENK with a cut-off value set at 57.3 pmol/L was characterized by a moderate sensitivity and specificity of 0.69 and 0.76, respectively, in AKI diagnosis in critically ill patients [110]. Furthermore, numerous studies reported that PENK is not impacted by inflammation and thereby, unlike NGAL, effectively predict AKI development in septic patients and 30-day mortality associated with kidney failure [54,55,111,112]. These results suggest that PENK is a reliable predictor of early AKI in ICU patients and improves overall recovery and survival. Moreover, a post-hoc analysis from the randomized ELAIN clinical trial showed that low baseline serum PENK concentrations are associated with an early and successful liberation from RRT. In contrast, higher levels are correlated with worse prognosis in patients with RRT-associated AKI [113]. Despite the results supporting the hypothesis that PENK may be another promising biomarker of AKI, relatively independent from many background factors such as inflammation, muscle mass, or age, there are still questions regarding optimal cut-off values and lack of consistency in different clinical settings. Therefore, future studies should be focused on clarifying those limitations.

### 3.6. Liver-Type Fatty Acid-Binding Protein (L-FABP)

L-FABP is a protein representing a family of lipid-binding proteins encoded by the *FABP1* gene in humans. It is involved in the uptake and intercellular transport of free fatty acids and their β-oxidation. As the name suggests, it is widely expressed in the liver but also in stomach, intestine, lungs, and kidney [114]. L-FABP can be classified as a marker of tubular injury as its expression is upregulated in the proximal convoluted tubule of the nephron in response to different inflammatory states such as ischemia-reperfusion injury and may be related to tubulointerstitial damage [115,116]. A recent systematic review and meta-analysis by Wilnes et al. of nine studies showed that urinary L-FABP levels were significantly elevated in children with AKI who underwent cardiopulmonary bypass and were linked with a worse condition and prognosis as well as a longer duration of hospital stay. The diagnostic performance of AKI was determined by an AUC value of 0.77 [117]. Other studies also showed L-FABP as an effective predictor of AKI in different clinical settings [61,62,118,119]. Furthermore, the study conducted by Pan et al., which involved critically ill patients admitted to the ICU, reported that urinary L-FABP was characterized by the highest predictive performance among other biomarkers, including NGAL, KIM-1, and interleukin-18 for mortality, and a lower concentration of this biomarker was associated with better prognosis and earlier liberation from RRT [120]. L-FABP also outperformed TIMP2*IGFBP7 in comparing the diagnostic efficacy of AKI after emergency laparotomy. The concentrations of L-FABP increased significantly at all time points, whereas TIMP2*IGFBP7 levels did not change significantly [117]. The significant limitation of L-FABP is its strong correlation with anemia and albuminuria. These symptoms are often present in patients with diagnosed pre-renal and intrinsic renal AKI, respectively; therefore, the diagnostic useability of L-FABP in AKI detection and discrimination may be significantly affected [121].

## 4. Novel Biomarkers in AKI Prediction and Diagnosis in Liver Transplant Recipients

Nowadays, liver transplantation (LTx) constitutes a routinely performed surgical procedure worldwide, being a definitive and radical treatment for end-stage liver disease [122]. There are two basic LTx types: orthotopic liver transplantation (OLTx) and heterotopic liver transplantation (HLTx). OLTx is a treatment of choice in most clinical situations when symptoms of decompensated cirrhosis or acute liver failure have occurred. It refers to a situation where a new healthy liver is transplanted in the place of a failing one. HLTx is an alternative technique, far less common than OLTx, where the new organ is transplanted, and the native liver remains in the recipient’s body. OLTx is associated with significantly better long-term graft function and is much more commonly performed [123]. Over the last decade, LTx etiologies have changed significantly in the United States and Europe. Currently, alcohol-associated liver disease (ALD) and metabolic dysfunction-associated fatty liver disease (MAFLD) are the most common indications for LTx, with a rapidly increasing prevalence in developed countries [124,125]. Because of enhanced surgery techniques, immunosuppressive therapy, and novel imaging methods, the post-transplant long-term outcomes and survival have significantly improved over the last few years. However, LTx is still burdened with various complications, including vascular and thrombotic events, biliary tract obstruction, abdominal infections, and AKI [126]. AKI associated with liver transplantation, often referred to as post-liver transplantation AKI, is one of the most common complications with a significant impact on morbidity and mortality [1]. According to the data, approximately 50% of patients undergoing LTx developed AKI, which was correlated with a significantly reduced survival rate in the Kaplan−Meier analysis. The prognosis was worsened when the need for RRT occurred [127]. The risk factors of post-liver transplantation AKI development include inter-alia, elderly, hypovolemia, electrolyte/acid-base balance disorders, duration of the anhepatic phase, ischemia-reperfusion injury, as well as pre-existing CKD [128,129]. As was mentioned above, the current diagnostic KDIGO criteria based on sCr concentrations and decrease in urine output may lead to the delayed diagnosis of post-transplant AKI as the early-stage alterations in sCr may not be detectable (especially <48 h from the beginning of injury) [7]. Therefore, sCr as a marker of AKI could be supported with novel molecules to enhance the diagnosis of AKI in liver recipients. Some researchers reported a possible role of different AKI biomarkers in the notably improved prediction and diagnosis of post-liver transplant AKI in comparison with sCr and urine output-based AKI criteria alone. The designs and outcomes of these studies are presented in Table 2. The most widely studied biomarker of AKI in LTx so far is NGAL. The available data showed serum and urinary NGAL as a good predictor of post-transplant AKI that allows for its discrimination, evaluation of perioperative mortality, and, in some settings, also a need for RRT [35,130,131,132,133,134]. On the contrary, in the prospective analysis of 92 patients undergoing orthotopic liver transplantation, urinary NGAL was not a reliable biomarker of AKI with a relatively low AUC value at the selected cut-off expressed as the urinary NGAL and urinary creatinine ratio [135]. In their observational study, Portal et al. showed that serum and urinary NGAL effectively predicted severe AKI early after the procedure and outperformed sCr-based diagnostic criteria. In a direct comparison, the AUC value of NGAL was 0.87, whereas sCr was 0.81 at day 0 [136]. Noteworthy, most of the current studies had a relatively small sample size that often did not exceed 50 participants. Discrepancies in AUC and cut-off values may lead to inconsistent outcomes, different sensitivity or specificity, and subsequently a significantly reduced diagnostic value of NGAL. Interestingly, some data report that modifications to NGAL by adding other clinical variables may be associated with improved results. For instance, Cho et al. showed that lactate-adjusted NGAL is a better predictor of early allograft dysfunction and AKI than NGAL or lactate alone after liver transplantation. This again suggests combining dynamic changes in serum or urinary NGAL with other laboratory parameters and scores may improve diagnostic efficacy [137]. Other biomarkers were far less studied than NGAL. A recent cohort study on 57 patients undergoing LTx conducted by Lima et al. revealed that serum PENK was an effective independent biomarker of the severe post-transplantation AKI 48 h after the procedure with a high AUC, specificity, and sensitivity at the selected cut-off [17]. Promising results were also obtained for urinary L-FABP in an observational study conducted on 27 liver transplant recipients. In this setting, L-FABP predictive value was evaluated dynamically at different time intervals during transplantation. The diagnostic performance of this biomarker was the highest at 4 h after the anhepatic phase in patients who developed postoperative AKI during 14-day follow-up [138]. The predictive value in post-transplant AKI of previously discussed urinary TIMP2*IGFBP7 was evaluated in two different studies. This biomarker did not efficiently predict mild/moderate and severe AKI with a low AUC value and did not show the same results as NGAL and KIM-1. However, both studies included a relatively small group of patients [130,139]. A clinical trial conducted by Sirota et al. on 40 LTx recipients assessed and compared different biomarkers in AKI prediction. Among tested molecules, concentrations of urinary NGAL, serum, and urinary interleukin 8, as well as urinary interleukin 18, were increased substantially in patients who developed AKI after LTx. These alterations were reflected in satisfying AUC values. In contrast, serum concentrations of CysC were not significantly different in patients with diagnosed AKI in comparison with those who did not develop this complication [140]. Furthermore, a recent prospective cohort study by Lima et al. compared the diagnostic performance of 19 biomarkers in severe AKI diagnosis and the need for RRT within the first week after LT using classification and regression tree cross-validation. In this study, urinary NGAL and L-FABP showed the highest efficacy in predicting the need for RRT during the postoperative period. In contrast, urinary KIM-1 and urinary osmolality outperformed other biomarkers in severe AKI diagnosis. An important strength of this clinical trial was a comprehensive analysis of biomarkers related to different aspects of kidney function, including injury, secretion, filtration, and fibrosis during the LT setting [141]. Undoubtedly, some of the novel biomarkers may be useful in predicting post-liver transplant AKI. A combination of widely used KDIGO criteria based on sCr and urine output alterations, clinical classifications like AKIN and RIFLE, and novel biomarkers may lead to the improved early diagnosis of AKI, thereby allowing for prompt intervention. Considering the significant burden of AKI on mortality and morbidity in the perioperative period in liver transplant recipients, this approach may contribute to the considerably increased overall transplantation success rate and survival. Nevertheless, studies on novel AKI biomarkers in LTx remain scarce; therefore, presented here, available data should be fulfilled by further investigations.

## 5. Novel Biomarkers in AKI Prediction and Diagnosis in Kidney Transplant Recipients

The prevalence of CKD is increasing rapidly worldwide. CKD constitutes an enormous socioeconomic burden for healthcare providers with parallelly growing costs along with the disease progression [144]. Currently, about 10% of the global population suffer from CKD, and it is estimated that this number will exceed 16.5% by 2032. This hazardous trend is correlated with a marked increase in years of life lost due to CKD, making it one of the leading causes of premature deaths [145]. The main risk factors for CKD development are elderly, diabetes mellitus, and cardiovascular disorders, especially hypertension [146]. Prevention and early diagnosis of CKD to slow down its progression into the end-stage kidney disease (ESKD) are the most crucial concerns. Kidney transplantation (KTx) is a treatment of choice and the gold standard for ESKD. Nowadays, KTx has become the world’s most performed solid organ transplantation. Despite continuous improvement in surgery techniques and perioperative care allowing for a substantial increase in survival and preservation of the graft function throughout the years, serious complications may impact the outcomes of KTx recipients [147]. Renal artery and vein thrombosis, transplant artery stenosis, arterial-venous fistulas, ureteral stenosis, allograft rejection, and AKI in transplanted kidneys are considered some of the most clinically important complications that may significantly affect graft function [148]. AKI constitutes a common finding in both kidney donors and recipients that may significantly affect short- and long-term transplantation outcomes. Post-kidney transplant AKI manifests as delayed graft function (DGF) or acute graft failure and is often linked with the necessity for RRT shortly after surgery. Older age, higher sCr levels after KTx, grafts coming from donation after circulatory death, and high immunological risk profiles are independent factors of AKI development [4,149]. An early diagnosis of post-kidney transplant AKI, especially in vulnerable individuals, is a pivotal step in undisturbed graft condition and function. Many AKI biomarker studies show their potential in improved and accelerated post-kidney transplant AKI diagnosis. However, most of these data are single-center studies with a relatively small number of participants. A recent meta-analysis conducted by Marc et al. revealed that many studies evaluating the diagnostic performance of AKI biomarkers in KTx had serious pitfalls, namely, the predomination of retrospective analysis, beautification of the data, lack of transparency, accurate interpretation of obtained results, and insufficient validation [150]. Therefore, future research on biomarkers should be more rigorous, focused on larger sample sizes, clearly defined study groups, and accurate statistical analysis. We believe these comments do not diminish the scientific value of the currently published data. They rather underline important concerns to improve the quality of future studies. In Table 3, we showed available clinical studies on AKI biomarkers in the KTx with the received outcomes. AKI as a complication of KTx was significantly less intensively studied than in LTx. A body of evidence examined urinary NGAL diagnostic utility in diagnosing AKI under the form of DGF or acute graft rejection probability occurrence. For instance, Koo et al. reported that urinary NGAL fairly predicted an incidence of RGF after KTx with good discrimination between patients who did not develop it. However, the coincidence interval was relatively wide. Furthermore, urinary NGAL outperformed significantly both urinary KIM-1 and L-FABP in this setting [14]. On the contrary, other authors revealed that the diagnostic performance of urinary biomarkers was limited, especially in DGF prognosis and assessment of the early-graft function after transplantation, which is based mainly on eGFR monitoring [20,151,152]. Ramirez-Sandoval et al. analyzed the diagnostic value of urinary NGAL in the same group of kidney recipients. NGAL was a robust predictor of AKI development after KTx during 1-year follow-up, whereas its effectiveness in immunological acute graft rejection shortly after surgery was below expectations [152,153]. In patients after living- or deceased-donor KTx who developed acute graft dysfunction or RGF that required intervention via RRT, TIMP2*IGFBP7 concentrations were significantly elevated and correlated with decent AUC values in comparison with those in patients who did not present these complications [154,155]. Despite the constant and rapid evolvement of the novel predictive biomarkers field, their diagnostic performance in KTx should be further investigated. Avoiding the common pitfalls in future research methodologies and designs is required to provide reliable results that could contribute to the wider adoption of AKI biomarkers into clinical use. The potential role of metabolomic or genomic profiling and identification in AKI diagnosis after KTx has been studied recently. Zhang et al. revealed that in kidney recipients who developed AKI after transplantation, serum metabolic profiling with usage of ultra-high-performance liquid chromatography-tandem mass spectrometry was associated with disturbances in tryptophan and arginine pathways during follow-up. Concentrations of these two amino acids were significantly lower than in healthy controls. Moreover, altered circulating forms of tryptophan and symmetric dimethylarginine were detected and their combination was characterized by a high AUC value, thereby outperforming sCr [156]. Another recent study showed that analysis of the renal metabolome may be a useful tool in prediction of post-transplantation AKI, as well as long-term graft survival. In this study, authors created a model basing on K-folded cross-validation of different compound combinations that predicted one-year eGFR effectively. The material was analyzed from biopsies of the grafts collected during the reperfusion phase of the surgery. Increased abundance of glucose-1-phosphate and fumarate was positively correlated with one-year eGFR, whereas higher levels of succinate and arginosuccinate were correlated negatively and associated with worse outcomes [157]. In turn, the bioinformatic analysis performed by Kang et al. showed that altered expression of selected hub genes involved in renal epithelial cell proliferation is associated with higher risk of post-transplantation AKI development, thereby pointing them out as a potential genetic biomarkers [158]. Another recent bioinformatics study allowed for the identification of six ferroptosis-related hub genes involved in ischemia-reperfusion injury development during or after KTx. Furthermore, the blockade one of them, namely, CD44 in a mouse model by specific antibodies, led to the inhibition of ferroptosis and accumulation of macrophages which was reflected by a substantially decreased renal damage score caused by often inevitable ischemia-reperfusion injury [159]. In addition, monitoring of nucleic acids levels in kidney biopsies such as cell-free DNA or microRNA seem to be a valuable diagnostic approach toward post-transplantation AKI [160,161]. Considering that various metabolic pathways and genes are involved in AKI pathogenesis, profiling and identification of both kidney metabolome or genome may constitute another step forward in developing new AKI biomarkers. However, further studies in kidney transplantation settings are needed as the current data in this area are very limited.

## 6. Conclusions

There is no doubt that novel predictive biomarkers shed new light on AKI prediction and diagnosis in kidney or liver transplant recipients. Preoperative selection of susceptible individuals followed by early recognition of AKI is crucial for transplantation success, reduction of the graft rejection events, and, subsequently, improved patient survival. Noteworthy, the goal of this review was not to advocate replacing the most widely used, standardized, and cost-effective sCr-based calculations with biomarkers, but rather to highlight their supportive role in early AKI diagnosis. Combining KDIGO criteria with biomarkers in multiplex diagnostic panels as well as metabolomic and genomic profiling could be an essential game-changer in managing post-transplantation AKI. Future research perspectives should consider multi-center, prospective studies with a larger number of participants, as well as rigorous and accurate data interpretation with clearly defined outcomes. Considering the rapidly growing prevalence of chronic kidney and liver diseases in the worldwide population as well as the increasing population of potential transplant candidates, we believe that this review, at least partially, will encourage researchers to investigate further AKI biomarkers, especially in the clinical setting of liver or kidney transplantation where the data remain scarce.

## Figures and Tables

**Figure 1 ijms-25-12072-f001:**
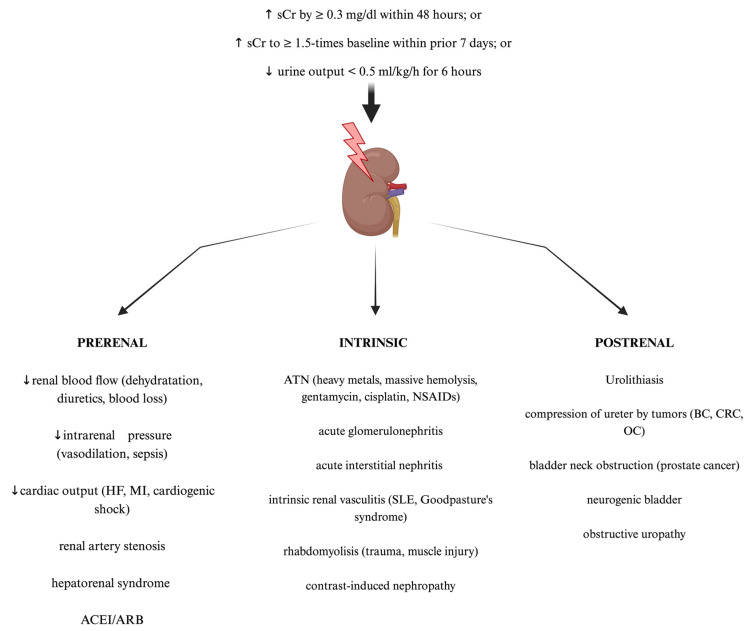
**The standard classification and most common causes of acute kidney injury (AKI). Created with Biorender.com.** ACEI—angiotensin converting enzyme inhibitors; ARB—angiotensin receptor blockers; ATN—acute tubular necrosis; BC—bladder cancer; CRC—colorectal cancer; HF—heart failure; MI—myocardial infarction; NSAIDs—non-steroid anti-inflammatory drugs; OC—ovarian cancer; sCr—serum creatinine; SLE—systemic lupus erythematosus.

**Figure 2 ijms-25-12072-f002:**
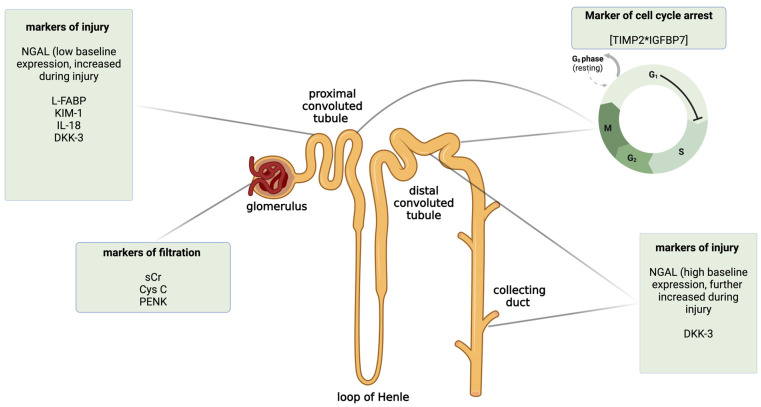
The classification of selected novel acute kidney injury biomarkers by the mechanism of action and the predominant place of expression in the nephron. Created with Biorender.com. CysC—cystatin C; DKK-3—dickkopf 3; IL-18—interleukin 18; KIM-1—kidney injury molecule 1; L-FABP—liver-type fatty acid-binding protein; NGAL—neutrophil gelatinase-associated lipocalin; PENK—proenkephalin; sCr—serum creatinine; TIMP2*IGFBP7—insulin-like growth factor-binding protein 7 and tissue inhibitor metalloproteinase 2.

**Table 1 ijms-25-12072-t001:** A summary of the selected clinical studies on novel predictive biomarkers in AKI prediction and diagnosis in different clinical settings other than liver or kidney transplantation.

Biomarker	Clinical Setting	Outcomes	Reference
**serum NGAL**	ICU, *n* = 138	AUC: 0.80 for AKI diagnosis cut-off: 391 ng/mL	[29]
	ICU, *n* = 94	mean AUC: 0.723 for AKI diagnosis cut-off: 150 ng/mL sen.: 85.3%; spe.: 50%	[30]
	myocardial infarction *n* = 385	AUC: 0.943 for AKI diagnosis cut-off: 112.5 ng/mL sen.: 86.4%; spe.: 100%	[31]
**urinary NGAL**	HF, *n* = 100	AUC: 0.778 for AKI diagnosis cut-off: 12 ng/mL sen.: 79%; spe.: 67%	[32]
	HF, *n* = 72	AUC: 0.84 for AKI diagnosis sen.: 74.7%; spe.: 84%	[33]
	coronary angiography *n* = 30	AUC: 0.68 for AKI diagnosis NGAL/creatinine ratio cut-off: 56.4 μg/mg creatinine sen.: 82.6%; spe.: 53.3%	[34]
	decompensated cirrhosis *n* = 213	C-statistic for AKI discrimination between ATN-AKI and non-ATN-AKI: 0.762 cut-off: 244 μg/g creatinine; sen.: 71%; spe.: 76% C-statistic for 90-day transplant-free survival NGAL vs. MELD-Na: 0.697 vs. 0.683, respectively	[35]
**serum CysC**	cirrhosis, *n* = 540	AUC: 0.940 for AKI diagnosis cut-off: 1.24 mg/L sen.: 92%; spe.: 92%	[36]
	cirrhosis, *n* = 100	AUC: 0.832 for AKI diagnosis cut-off: 0.94 mg/L sen.: 75.6%; spe.: 89.8%	[37]
	acute liver failure, *n* = 56	AUC: 0.975 for AKI diagnosis cut-off: 1.21 mg/L sen.: 100%; spe.: 87.5	[38]
	ICU, *n* = 414	AUC: 0.842 for AKI diagnosis cut-off: 1.25 mg/L sen.: 82.2%; spe.: 76.4	[39]
	acute pancreatitis, *n* = 237	AUC: 0.948 for AKI diagnosis cut-off: 1.865 mg/L sen.: 88.9%; spe.: 100%	[40]
	acute pancreatitis, *n* = 379	AUC: 0.711 for AKI diagnosis cut-off: 1.055 mg/L sen.: 45.5%; spe.: 86.7	[41]
**urinary KIM-1**	meta-analysis, *n* = 42	AUC: 0.907 for AKI diagnosis sen.: 0.86; spe.: 0.84	[42]
	methotrexate or platinum-based antineoplastic therapy, *n* = 64	AUC: 0.82 for AKI diagnosis cut-off: 6.2 ng/mg of creatinine sen.: 73.1%; spe.: 92.1%	[43]
	cirrhosis, *n* = 150	AUC: 0.843 for AKI diagnosis cut-off: 4.56 ng/mL sen.: 77.2%; spe.: 79.8%	[44]
	HIV, *n* = 468	Higher levels of KIM-1 were significantly associated with an increased risk of AKI development (HR: 1.30; 95% CI: 1.03–1.63)	[45]
	ICU, *n* = 86	AUC: 0.81 for AKI diagnosis cut-off: 0.8 ng/mg of creatinine sen.: 91%; spec.: 81%	[46]
**urinary TIMP2*IGFBP7**	cardiac surgery postoperatively, *n* = 100	AUC: 0.541 for stage 1–2 AKI AUC: 0.838 for stage 3 AKI	[47]
	cardiac arrest, *n* = 115	AUC: 0.91 for severe AKI diagnosis cut-off: 0.39 (ng/mL)^2^/1000 sen.: 97%; spec.: 72%	[48]
	ICU, *n* = 728	AUC: 0.800 for moderate/severe AKI diagnosis	[49]
	ICU, *n* = 209	AUC: 0.774 for AKI diagnosis cut-off: 0.81 (ng/mL)^2^/1000 sen.: 62%; spe.: 83.8%	[50]
	emergency department, *n* = 368	C-statistic for AKI diagnosis: 0.54	[51]
	ICU, *n* = 100	AUC: 0.66 for AKI diagnosis alone cut-off: 2.63 (ng/mL)^2^/1000 sen.: 61%; spe.: 71% Parameters were improved when combined with FST.	[52]
	cisplatin antineoplastic therapy, *n* = 156	AUC range: 0.61–0.7 for AKI diagnosis	[53]
**serum PENK**	ICU, *n* = 150	28-day mortality: HR, 0.785; 95% CI, 0.706–0.865 cut-off: 0.36 ng/μL	[54]
	ICU, *n* = 167	AUC: 0.725 for AKI diagnosis cut-off: 154 pmol/L sen.: 65.9%; spe.: 79.4% 30-day mortality: HR, 7.9 95%; CI, 3.9–16.2 cut-off: 80 pmol/L	[55]
	CKD, *n* = 111	AUC: 0.92 for AKI diagnosis	[56]
	Thoraco-abdominal aortic aneurysm, *n* = 33	AUC: 0.866 for AKI diagnosis	[57]
**urinary dickkopf-3**	ICU, *n* = 420	AUC: 0.80 for AKI diagnosis AUC: 0.78 for mortality	[58]
	coronary angiography, *n* = 490	AUC: 0.61 for AKI diagnosis sen.: 47.4%; spe.: 72.4	[59]
	CVD, *n* = 2344	HR: 1.07; 95% CI: 0.85–1.21 for risk of AKI	[60]
**urinary L-FABP**	emergency laparotomy *n* = 48	AUC: 0.8 for AKI diagnosis sen.: 55.6%; spe.: 91.9%	[61]
	ICU, *n* = 152	AUC: 0.79 for AKI diagnosis	[62]
	HF, *n* = 281	AUC: 0.930 for AKI diagnosis cut-off: 12.5 μg/g of creatinine sen.: 94.2%; spe.: 87%	[63]
	coronary angiography, *n* = 193	AUC: 0.642 for AKI diagnosis cut-off: 20.7 ng/mg of creatinine sen.: 54%; sen.: 62%	[64]
	trauma, *n* = 100	OR: 18.24 95% CI: 4.21–79.02 for AKI diagnosis sen.: 73.3%; spe.: 88.2%	[65]
**urinary glycine and ethanolamine**	ICU, *n* = 121 metabolomic profiling	decreased concentrations of glycine and ethanolamine in AKI patients	[66]
**serum phenylalanine**	sepsis, *n* = 63 metabolomic profiling	AUC: 0.89 for AKI diagnosis within 24 h after admission to the ICU intensified metabolism of phenylalanine pathaways in sepsis-associated AKI	[67]
**serum oxidized lipid metabolites**	sepsis, *n* = 67 metabolomic profiling	altered metabolism of 21 oxidized lipid metabolites in patients with sepsis-associated AKI	[68]
**serum I3A**	cardiac surgery, *n* = 55 metabolomic profiling	increased serum concentration of I3A in patients with AKI AUC: 0.84 for AKI diagnosis perioperatively and intraoperatively	[69]
**urinary metabolite panel:** **Tyr-gGlu + DAGC + Arg-arg + L-Met + AAMU**	coronary artery bypass graft, *n* = 55 metabolomic profiling	AUC: 0.89 for AKI diagnosis postoperatively sen.: 86%; spe.: 74%	[70]
**serum 5-HIAA**	vancomycin-associated AKI, *n* = 28 metabolomic profiling	increased concentration of 5-HIAA in AKI patients increased 5-HIAA/5-HT ratio in AKI patients AUC: 0.795 for 5-HIAA for AKI diagnosis AUC: 0.884 for the 5-HIAA/5-HT ratio for AKI diagnosis	[71]

95% CI—95% confidence interval; 5-HIAA—5-hydroxyindoloacetic acid; 5-HT—serotonin; AAMU—5-acetyloamino-6-amino-3-methylurcil; AKI—acute kidney injury; Arg-arg—arginyl-arginine; AUC—area under the curve; CVD—cardiovascular disease; CysC—cystatin C; DAGC—deoxycholic acid glycine conjugate; DKK-3—dickkopf 3; FST—furosemide stress test; HF—heart failure; HIV—human immunodeficiency virus; HR—hazard ratio; I3A—indole-3-aldehyde; ICU—intensive care unit; KIM-1—kidney injury molecule 1; L-FABP—liver-type fatty acid-binding protein; L-met—L-methionine; MELD-Na—model for end-stage liver disease score with sodium; NGAL—neutrophil gelatinase-associated lipocalin; OR—odds ratio; PENK—proenkephalin; sen.—sensitivity; spe.—specificity; TIMP2*IGFBP7—insulin-like growth factor-binding protein 7 and tissue inhibitor metalloproteinase 2; Tyr-gGlu—tyrosyl-gamma-glutamate. All presented results were considered as statistically significant (*p* < 0.05).

**Table 2 ijms-25-12072-t002:** A summary of the clinical studies on novel predictive biomarkers in AKI prediction and diagnosis after liver transplantation.

Biomarker	Clinical Setting	Outcomes	Reference
**serum NGAL**	LD-LTx, *n* = 353	NGAL alone: AUC: 0.74 for AKI diagnosis; OR: 0.84 NGAL adjusted with lactate: AUC: 0.9 for AKI diagnosis; OR: 0.89	[137]
	LTx, *n* = 100	2-fold higer NGAL levels in patients with severe AKI 18 h after transplantation cut-off: 198 ng/mL sen.: 87%; spe.: 71%	[131]
	LTx, *n* = 26	AUC: 0.86 for severe AKI diagnosis 8 h after transplantation cut-off: 243.5 ng/mL	[134]
	LTx, *n* = 95	AUC: 0.87 for severe AKI diagnosis 12 h after transplantation cut-off: 258 ng/mL	[136]
**urinary NGAL**	LTx, *n* = 16	AUC: 0.816 for AKI diagnosis 24 h after transplantation	[130]
	LTx, *n* = 100	AUC: 0.76 for severe AKI diagnosis 6 h after transplantation cut-off: 136 ng/mL sen.: 68%; spe.: 76%	[131]
	LTx, *n* = 27	AUC: 0.792 for AKI diagnosis 24 h after transplantation AUC: 0.812 for predicting the need for RRT 24 h after transplantation	[132]
	LTx, *n* = 26	AUC: 0.76 for severe AKI diagnosis 8 h after transplantation cut-off: 94.5 ng/mL	[134]
	OLTx, *n* = 45	AUC: 0.79 for AKI diagnosis 24 h after transplantation	[133]
	LTx, *n* = 92	AUC: 0.636 for AKI diagnosis 18 h after transplantation cut-off: 35 ng/mg of urinary creatinine sen.: 68.1%; spe.: 59.7%	[135]
	LTx, *n* = 95	AUC: 0.8 for severe AKI diagnosis 12 h after transplantation cut-off: 258 ng/mL	[136]
	LTx, *n* = 100	AUC: 0.62 for severe AKI diagnosis in the first week after transplantation	[141]
**urinary TIMP2*IGFBP7**	LTx, *n* = 16	AUC: 0.683 for AKI diagnosis 24 h after transplantation	[130]
	OLTx, *n* = 40	AUC: 0.71 for severe AKI diagnosis 48 h after transplantation	[139]
**urinary KIM-1**	LTx, *n* = 16	AUC: 0.9 for AKI diagnosis 24 h after transplantation	[130]
**serum PENK**	LTx, *n* = 57	preoperatively: AUC, 0.69 for severe AKI diagnosis; cut-off, 55.3 pmol/L; sen., 86%; spe., 0.52 postoperatively: AUC, 0.83 for severe AKI diagnosis; cut-off, 119.05 pmol/L; sen., 81%; spe., 90%	[17]
	LTx, *n* = 100	AUC: 0.7 for severe AKI diagnosis in the first week after transplantation.	[141]
**serum ATIII**	LD-LTx, *n* = 577	AUC: 0.709 for AKI diagnosis sen.: 71.3%; spe.: 64.1% OR: 2.839 95% CI: 1.311–6.147 for a 2.8-fold higher AKI probability for low ATIII levels	[142]
**serum renalase**	LD-LTx, *n* = 50	AUC: 0.54 for AKI diagnosis	[12]
**serum L-FABP**	LTx, *n* = 25	AUC: 0.760 for AKI diagnosis 4 h after transplantation cut-off: 3451.75 ng/mg of urinary creatinine sen.: 72.7%; spe.: 71.4%	[138]
**urinary de novo NAD^+^**	OLTx, *n* = 49 metabolomic profiling	AUC: 0.729 for AKI diagnosis	[143]

95% CI—95% confidence interval; AKI—acute kidney injury; ATIII—antithrombin III; AUC—area under the curve; h—hours; KIM-1—kidney injury molecule 1; LD-LTx—living-donor liver transplantation; L-FABP—liver-type fatty acid-binding protein; LTx—liver transplantation; *n*—sample size; NAD^+^—nicotinamide adenine dinucleotide; NGAL—neutrophil gelatinase-associated lipocalin; OLTx—ortothopic liver transplantation; OR—odds ratio; PENK—proenkephalin; sen.—sensitivity; spe.—specificity; TIMP2*IGFBP7—insulin-like growth factor-binding protein 7 and tissue inhibitor metalloproteinase 2. All presented results were considered as statistically significant (*p* < 0.05).

**Table 3 ijms-25-12072-t003:** A summary of the clinical studies on novel predictive biomarkers in AKI prediction and diagnosis after kidney transplantation.

Biomarker	Clinical Setting	Outcomes	Reference
**serum NGAL**	KTx, *n* = 37	AUC: 0.83 for AKI diagnosis 7 days after transplantation cut-off: 1.6 mg/dL sen.: 88.9%; spe.: 81%	[162]
**urinary NGAL**	KTx, *n* = 15	no significant diffrences in NGAL concentrations negative correlation between eGFR and NGAL (r = −0.77)	[151]
	KTx, *n* = 109	AUC: 0.758 for RGF diagnosis (95% CI: 0.645–0.871) OR for RGF: 2.143 95% CI: 0.920–4.992	[14]
	KTx, *n* = 67	AUC: 0.89 for AKI diagnosis during 1-year follow-up 95% CI: 0.81–0.97 cut-off: 200 ng/mL sen.: 84%; spe.: 86%	[153]
	KTx, *n* = 67	AUC: 0.89 for AKI diagnosis during 1-year follow-up 95% CI: 0.81–0.97 cut-off: 200 ng/mL sen.: 84%; spe.: 86%	[152]
**urinary TIMP2*IGFBP7**	LD-KTx, *n* = 48	AUC: 0.939 for acute allograft dysfunction diagnosis cut-off: 0.803 (ng/mL)^2^/1000 sen.: 94.4%; spe.: 83.3%	[155]
	DD-KTx, *n* = 56	AUC: 0.76 for RGF diagnosis 4 h after kidney reperfusion 95% CI: 0.62–0.91	[154]
**urinary KIM-1**	DD-KTx, *n* = 109	AUC: 0.506 for RGF diagnosis 95% CI: 0.391–0.620	[14]
**urinary L-FABP**	KTx, *n* = 109	AUC: 0.704 for RGF diagnosis 95% CI: 0.592–0.817	[14]
**urinary DKK3**	KTx, *n* = 122	OR for RGF diagnosis: 4.001 95% CI: 0.994–16.100 Higher concentrations of DKK-3 measured three and twelve months after transplantation predicted AKI.	[18]
**serum tryptophan and SDMA**	KTx, *n* = 42 metabolomic profiling	altered metabolism and decreased concentrations of tryptophan and SDMA AUC: 0.900 for AKI diagnosis for combined tryptophan and SDMA AUC: 0.820 for AKI diagnosis for SDMA alone AUC: 0.738 for AKI diagnosis for tryptophan alone	[156]
**G1P, fumarate, and succinate**	KTx biopsies, *n* = 42 metabolomic profiling	One-year eGFR positively correlated with the increased abundance of G1P and fumarate. One-year eGFR negatively correlated with the increased abundance of succinate.	[157]
**Hub genes involved in renal cell proliferation** **(*AKAP12*, *AMOT*, *C3AR1*, *LY96*, *PLCD4*, *PLCG2*, and others)**	bioinformatics analysis of gene expression from the Omnibus database genomic profiling	Altered expression of selected hub genes was associated with a higher risk of post-transplantation AKI development.	[158]
**dd-cfDNA**	KTx biopsies, *n* = 604 genomic profiling	elevated serum level of dd-cfDNA during post-transplantation AKI	[160]
**microRNA (miR-182-5p)**	KTx biopsies, *n* = 166 genomic profiling	miR-182-5p expression correlated significantly with genes involved in AKI development.	[161]

95% CI—95% confidence interval; AKI—acute kidney injury; AUC—area under the curve; dd-cfDNA—donor-derived cell-free deoxyribonucleic acid; DD-KTx—deceased-donor kidney transplantation; DKK-3—dickkopf 3; G1P—glucose-1-phosphate; KIM-1—kidney injury molecule 1; LD-KTx—living-donor kidney transplantation; L-FABP—liver-type fatty acid-binding protein; KTx—kidney transplantation; *n*—sample size; NGAL—neutrophil gelatinase-associated lipocalin; OR—odds ratio; RGF—reduced graft function; SDMA—symmetric dimethylarginine; sen.—sensitivity; spe.—specificity; TIMP2*IGFBP7—insulin-like growth factor-binding protein 7 and tissue inhibitor metalloproteinase 2. All presented results were considered as statistically significant (*p* < 0.05).

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
