# Peer review of "Can Novel Biomarkers Effectively Predict Acute Kidney Injury in Liver or Kidney Transplant Recipients?"

_ijms, 2024, doi:10.3390/ijms252212072_

Round 1

Reviewer 1 Report

Comments and Suggestions for Authors

The review is written well and nicely.

Some minor comments.

Numerous studies have pointed out NGAL as a promising predictive biomarker of AKI, especially in acute states.

Can the authors mention those studies?

For the AKI prediction, the authors mentioned literature on proteins. Is there also literature on metabolomics? If so, please mention them in a table.

In the following section, “Novel biomarkers in AKI prediction and diagnosis in kidney transplant recipients,” the authors mention only proteins. Please mention other studies involving metabolomics and genomics.

Author Response

Warsaw, Poland 04/11/2024

Dear Madam/Dear Sir,

We express our sincere thanks for the time and effort that you dedicated to reviewing our manuscript. We have been able to incorporate all the concerns related to the submitted manuscript. All the changes within the manuscript have been highlighted. Here are point-by-point responses to Reviewer comments:

The review is written well and nicely.

Authors: Thank you very much for your positive remarks on our article. We do appreciate it. It is a strong motivation for us in the future work in this area.

Some minor comments.

Numerous studies have pointed out NGAL as a promising predictive biomarker of AKI, especially in acute states.

Can the authors mention those studies?

Authors: Thank you for pointing this out. We have already mentioned a few studies showing NGAL as a promising predictive biomarker of AKI in acute states, namely, in patients admitted to intensive care units or percutaneous coronary intervention related to myocardial infarction.

However, considering the Reviewer’s comment, we think that this sentence may sound too general and wordy, so we decided to rebuild it to make it clearer, more cohesive, and easier to read, thereby avoiding misunderstandings. Furthermore, we added some studies focused on acute states, including sepsis, heart failure, and decompensated cirrhosis, supported by appropriate references (page no. 12, lines 190-194).

For the AKI prediction, the authors mentioned literature on proteins. Is there also literature on metabolomics? If so, please mention them in a table.

Authors: Studies on human metabolome were not the primary aim of this review. However, we entirely agree with the Reviewer that this area constitutes a promising approach toward developing new AKI biomarkers. This subject deserves another review article. Nevertheless, according to the Reviewer’s advice, we decided to include available clinical studies on alterations in human metabolome and their possible link with AKI diagnosis to highlight their importance in future research perspectives. We updated Table 1 according to the Reviewer’s instructions with appropriate references (Table 1).

In the following section, “Novel biomarkers in AKI prediction and diagnosis in kidney transplant recipients,” the authors mention only proteins. Please mention other studies involving metabolomics and genomics.

Authors: Thank you for your valuable suggestion. Both metabolome and genome seem promising yet uncharted areas in kidney transplantation research. We updated this section according to the Reviewer’s instructions reviewing available data (page no. 23, lines 537-561). Furthermore, we added available clinical studies on the human metabolome and genome to Table 3 (Table 3).

Considering all the Reviewer’s suggestions and concerns provided, we believe that the manuscript was significantly improved and is now more suitable for publication in the International Journal of Molecular Sciences.

Yours Faithfully,

Hubert Zywno

Department of Nephrology, Dialysis, and Internal Diseases

University Clinical Centre, Medical University of Warsaw

Banacha 1A st. 02-097 Warsaw

Poland

tel. +48225992768

Reviewer 2 Report

Comments and Suggestions for Authors

It’s a pleasure to review such an intriguing paper that provides valuable insights into the predictive accuracy of novel biomarkers for acute kidney injury in liver or kidney transplant recipients. The figures are well-crafted, engaging, and easy to understand, and the critical assessment of these new markers’ predictive effectiveness for early AKI diagnosis is excellent. However, the table design could benefit from some technical improvements.

Author Response

Warsaw, Poland 05/11/2024

Dear Madam/Dear Sir,

We express our sincere thanks for the time and effort that you dedicated to reviewing our manuscript. We have been able to incorporate all the concerns related to the submitted manuscript. All the changes within the manuscript have been highlighted.

It’s a pleasure to review such an intriguing paper that provides valuable insights into the predictive accuracy of novel biomarkers for acute kidney injury in liver or kidney transplant recipients. The figures are well-crafted, engaging, and easy to understand, and the critical assessment of these new markers’ predictive effectiveness for early AKI diagnosis is excellent. However, the table design could benefit from some technical improvements.

Authors: We want to extend our sincere gratitude for your generous and encouraging feedback on our manuscript. It is an honor to receive such a positive review, and we are truly appreciating your kind remarks. Your acknowledgment of our work on the predictive accuracy of novel biomarkers for acute kidney injury in transplant recipients is both humbling and motivating.

According to the Reviewer’s comment we entirely redesigned and formatted the tables with the aim to make them clearer and more readable (Table 1, 2, and 3).

Considering all the Reviewer’s suggestions and concerns provided, we believe that the manuscript was significantly improved and is now more suitable for publication in the International Journal of Molecular Sciences.

Yours Faithfully,

Hubert Zywno

Department of Nephrology, Dialysis, and Internal Diseases

University Clinical Centre, Medical University of Warsaw

Banacha 1A st. 02-097 Warsaw

Poland

tel. +48225992768
